# Radiation Therapy Changed the Second Malignancy Pattern in Rectal Cancer Survivors

**DOI:** 10.3390/medicina59081463

**Published:** 2023-08-15

**Authors:** Xiaoxian Ye, Yinuo Tan, Ruishuang Ma, Pengrong Lou, Ying Yuan

**Affiliations:** 1Department of Radiotherapy and Chemotherapy, The First Affiliated Hospital of Ningbo University, Ningbo 315000, China; 2Department of Medical Oncology (Key Laboratory of Cancer Prevention and Intervention, China National Ministry of Education, Key Laboratory of Molecular Biology in Medical Sciences), The Second Affiliated Hospital, Zhejiang University School of Medicine, Hangzhou 310009, China; 3Zhejiang Provincial Clinical Research Center for Cancer, Hangzhou 310058, China; 4Cancer Center of Zhejiang University, Hangzhou 310058, China

**Keywords:** rectal cancer, second malignancy, SEER, radiotherapy

## Abstract

*Background and Objectives:* Radiotherapy (RT) plays an important role in the treatment for locally advanced rectal cancer patients. It can bring radio exposure together with the survival benefit. Cancer survivors are generally at an increased risk for second malignancies, and survivors receiving RT may have higher risks than survivors not receiving RT. Whether the risk of an all-site second malignancy may increase after RT is still debated. This study aims to compare the second malignancy pattern in rectal cancer survivors after RT. *Materials and Methods:* The Surveillance, Epidemiology, and End Results (SEER) database was used for analysis. In total, 49,961 rectal cancer patients (20–84 years of age) were identified between 2000 and 2012 from 18 SEER registries. All patients underwent surgery. The occurrence of second malignancies diagnosed after rectal cancer diagnosis was compared in patients who received and did not receive RT. The standardized incidence ratios (SIRs) with 95% confidence intervals (CIs) were used. SEER*Stat was used to generate the 95% CIs for the SIR statistics using the exact method. *Results:* Of the total 49,961 patients, 5582 developed second malignancies. For all-site second primary malignancies, the age-adjusted SIRs were 1.14 (95% CI 1.1–1.18) and 1.00 (95% CI 0.96–1.04) in the no RT and RT groups, respectively. In 23,192 patients from the surgery-only group, 2604 had second malignancies, and in 26,769 patients who received RT, 2978 developed second malignancies. With respect to every site, the risk of secondary prostate cancer was significantly lower in the RT group (SIR = 0.39, 95% CI 0.33–0.46) than that in the surgery-only group (SIR = 1.04, 95% CI 0.96–1.12). Moreover, the risk of thyroid cancer was significantly higher in the RT group (SIR = 2.80, 95% CI 2.2–3.51) than that in the surgery-only group (SIR = 1.29, 95% CI 0.99–1.66). *Conclusions:* RT may change the second malignancy pattern in rectal cancer survivors; the risk of prostate cancer decreased, and the risk of thyroid cancer increased most significantly.

## 1. Introduction

Colorectal cancer is the fourth most common cancer worldwide [1]. In recent years, due to the active promotion of early screening and the improvement of clinical treatment, including surgery, radiotherapy (RT), chemotherapy, and immunotherapy, the survival outcome of patients with colorectal cancer has been significantly improved [1,2]. Second primary malignancy (SPM) is a multifactorial disease that is associated with heredity, treatment, lifestyle, and environmental factors. However, it is also considered as another risk factor for death following recurrence and metastasis [3]. RT plays an important role in the treatment of rectal cancer [4]. Studies have shown that RT can effectively reduce tumor recurrence and metastasis [5,6,7,8], but whether ionizing radiation increases the occurrence of other tumors is unclear.

Ionizing radiation is a potent carcinogen, inducing malignant tumors through DNA damage. Some studies found that the exposure to ionizing radiation may cause CML. Venkata S K Manem demonstrated that the increase in second cancer risks is directly correlated with increasing values of the linear energy transfer of charged particles (including protons, alpha particles, and heavy ions like carbon and neon). On the other hand, when undergoing rectal cancer radiotherapy, radiation may have a certain coverage of the prostate area. Thus, whether it may also have a certain impact on the incidence of prostate cancer has not been studied yet. Although several studies have investigated the association between RT and SPM in rectal cancer patients, the conclusions are different [9,10]. Here, we use the SEER database to compare the effects of radiation on the incidence of second primary tumors based on the large number of population studies.

## 2. Materials and Methods

### 2.1. Data Source

The cohort for this study was extracted from the Surveillance, Epidemiology, and End Results (SEER) database using the SEER dataset for 18 SEER registries from 2000 to 2012. The SEER data cover 27.8% of the total US population [11]. The demographic composition of the SEER registries and incidence trends from the SEER data are generally considered to be representative of the US population (according to the SEER*Stat software version 8.3.6). The multiple primary standardized incidence ratio (SIR), used to track the incidence of second malignancies, includes patient information from 18 registries (Detroit, Atlanta, Seattle (Puget Sound), San Francisco (Oakland), Utah, New Mexico, Connecticut, Iowa, and Hawaii).

### 2.2. Patients

The cohort included patients who were diagnosed with a first primary malignant rectal cancer (International Classification of Diseases for Oncology Third Edition histology classification codes 8140/3, 8144/2, 8144/3, 8210/2, 8210/3, 8211/2, 8211/3, 8213/2, 8213/3, 8220/2, 8220/3, 8221/2, 8221/3, 8261/3, 8262/2, 8263/3, 8480/2, 8480/3, 8481/2, 8481/3, 8490/2, and 8490/3) who were presented with their first cancer between 1 January 2000 and 31 December 2012. Patients were excluded if the first primary rectal cancer was stage IV. Patients were also excluded if they did not undergo surgery for rectal cancer.

### 2.3. Statistical Analyses

Second primary tumors are defined as tumors originating from other organs and tissues, and patients with second primary tumors must exclude concurrent primary tumors and ensure a minimum 2-month interval after diagnosis of primary cancer. Analyses were stratified by age at rectal cancer diagnosis (20–49 years, 50–59 years, 60–69 years, and 70–84 years), race (white, black, or other), and latency period (2–11 months, 12–59 months, 60–120 months, or >120 months from the date of rectal cancer diagnosis). Additionally, we evaluated the risk of second primary tumor and compared it to the incidence in the general population between patients who received and did not receive RT. SEER*Stat was used to generate the 95% confidence intervals (CIs) for SIR statistics using the exact method. An SIR is defined as the ratio of the number of observed cases divided by the number of expected cases. Categorical data were compared using chi-squared tests for nominal data and Jonckheere-Terpstra nonparametric tests for ordinal data. Univariate and multivariate logistic regression models were used to identify factors associated with SPM. Statistical significance was defined, and a two-sided *p* value <0.05 was considered statistically significant.

## 3. Results

### 3.1. Demographics of Patients with First Primary Rectal Cancer

The descriptive characteristics of first primary rectal cancer cases are summarized in Table 1. In total, 49,961 patients were selected through the selection procedure. Among these, 26,769 (54%) patients were treated with RT and surgery, and 23,192 (46%) patients only underwent surgery. The median ages were 61 and 67 years in the RT and no RT groups, respectively. Compared with the no RT group, more patients younger than 60 years of age received RT. Moreover, more patients received RT in 2008–2012 than in 2000–2004.

### 3.2. Second Cancer Site Analysis between the No Radiotherapy (RT) and RT Patients

The overall incidence of SPM was 11.2% (5582/49,961), 2604/26,769 in the RT group and 2978/23,192 in the no RT group (Table 2). For all-site SPM, the age-adjusted SIRs were 1.14 (95% CI 1.1–1.18) and 1.00 (95% CI 0.96–1.04) in the no RT and RT groups, respectively. For most-site SPM, the age-adjusted SIRs were similar between the RT group and the no RT group except for the male and female genital systems and thyroid. For prostate SPM, the age-adjusted SIRs were 0.96 (95% CI 0.87–1.05) and 0.38 (95% CI 0.33–0.44) in the no RT group and the RT group, respectively (*p* < 0.05) (Figure 1). Furthermore, for ovary SPM, the age-adjusted SIRs were 0.79 (95% CI 0.49–1.21) and 0.44 (95% CI 0.21–0.81) in the no RT group and the RT group, respectively (*p* < 0.05). Meanwhile, thyroid SPM increased significantly in the RT group, and the age-adjusted SIRs were 1.82 (95% CI 1.40–2.33) and 1.06 (95% CI 0.73–1.50) in the RT group and the no RT group, respectively. Moreover, for colorectal SPM, both the RT group and the no RT group had significantly increased age-adjusted SIRs (2.55 [95% CI 2.36–2.74] in the no RT group and 1.78 [95% CI 1.62–1.95] in the RT group).

### 3.3. Latency Period Analysis of Second Primary Malignancy between the No RT and RT Patients

Previously, we found that the incidence of second primary prostate cancer in patients with primary rectal cancer in the RT group was significantly lower than that in the no RT group (Table 3). However, the incidence of thyroid cancer was higher. Furthermore, we investigated the SPM between the RT group and the no RT group in different latency periods and found that the incidence of SPM was diverse in different latency periods (Table 4).

For all-site SPM, the median latency times from the first primary rectal cancer to the occurrence of the second malignancy of each type in the RT group and surgery-only group are all listed in Appendix A. The median latency time of the RT and surgery-only groups were 51.0 months. And the media latency time of prostate cancer in the RT group was obviously shortened compared to the surgery-only group. The age-adjusted SIRs in the 36–47-month latency were 1.3 (95% CI 1.2–1.45) and 0.83 (95% CI 0.72–0.94) in the no RT group and the RT group, respectively. Additionally, the age-adjusted SIRs in the 48–59-month latency were 1.19 (95% CI 1.06–1.33) and 0.84 (95% CI 0.73–0.96) in the no RT group and the RT group, respectively (Figure 2A). For rectal SPM, we found that in the 12–95-month latency, the no RT group had a higher SPM than the RT group (Figure 2C). However, in other latency periods, the SPM in the RT group and the no RT group was similar. For prostate cancer, the RT group had a significantly lower SPM than the no RT group in different latency periods, except for the 2–5-month latency. The biggest difference was observed in the 12–35-month latency (Figure 2B).

For thyroid SPM, the RT group had a significantly higher SPM than the no RT group mainly in the first 3 years after rectal cancer (2–35-month latency) (Figure 2D).

### 3.4. Clinical–Pathological Characteristics between Secondary Prostate Cancer of Rectal Cancer after RT or without RT

The prostate cancer SPM was significantly lower in the RT group than that in the no RT group, and in almost every latency period, we hypothesized that using RT for rectal cancer can incidentally treat the invisible prostate cancer. Thus, we tried to compare the clinical–pathological characteristics between secondary prostate cancer of rectal cancer after RT or without RT. The results show that secondary prostate cancer showed no difference when compared by age, race, and grade. Regarding the tumor stage, the no RT group had more localized stage secondary prostate cancer than the RT group, and the RT group had more regional and distant stage secondary prostate cancer than the no RT group.

## 4. Discussion

Overall, we observed that rectal cancer survivors had a different risk of developing a second primary cancer compared with the general US population. Currently, with more effective treatment of cancer, the second primary tumor in cancer survivors has become an urgent problem. Age, sex, and treatment methods are risk factors for its development. Additionally, RT has always been controversial on the treatment and development of malignant tumors [10,12]. It is often considered as an inducer of the second primary tumor. The purpose of using RT for rectal cancer is to reduce the risk of recurrence, which has been fully demonstrated [6,7]. In the past decade, encouraging results of preoperative RT have been observed, and preoperative RT has been frequently used in clinical practice [8,13,14]. However, studies investigating the risk of second primary cancer after rectal RT are insufficient.

In this study, we found that there were significant differences in the pattern of second primary cancer in patients who received RT or patients who did not receive RT. Specifically, patients who received RT had a lower risk of developing ovarian, prostate, and breast cancers and a higher risk of developing thyroid, lung, and bronchial cancers than patients who did not receive RT. The change in cancer risk in these patients is consistent with the results of previous studies. Lu M et al. did not find an increase in the risk of developing a second primary cancer in breast cancer patients after RT [15]. Warschkow et al. also used the SEER database to compare the difference of second primary cancer between colorectal cancer patients with and without RT. Similar to our results, they found a reduced risk of prostate cancer and an increased risk of endometrial cancer with RT [16]. Nevertheless, we found that the risk of thyroid cancer was increased in patients with RT as well, which was not observed in their results. In this study, we included people from the year 2000 onward, whereas they included people from 1973 onward. Hence, the population we included was more consistent than their population, and the RT techniques performed in our study were more advanced compared to the techniques performed in these previous studies.

Regarding the influence of RT on the second primary cancer, there are not only reports of rectal cancer, but also similar studies on other cancers. Jahreiß MC et al. found that the risk of second primary cancer in prostate cancer patients with external beam radiotherapy (EBRT) was increased and remained throughout the different EBRT eras [17]. A meta-analysis also indicated that prostate radiotherapy significantly increases the risk of subsequent rectal cancer [18]. Combined with our study results, we found that in a variety of cancers, the proportion of patients with second primary thyroid cancer is prone to increase after RT, which may be consistent with the current scientific reports that RT tends to increase the incidence of thyroid cancer. In our study, we found a significant reduction in the incidence of prostate cancer after RT, and we hypothesized that this might be due to the tumor cells in the prostate being vulnerable to death during rectal radiation. Of course, pelvic radiation can more or less affect the physiological function or cellular viability of the pelvic reproductive organs (the testicle and ovary), causing a decrease in androgen and/or estrogen levels [19,20,21], which may be the main factor for the occurrence and development of hormone-dependent tumors (such as prostate cancer and breast cancer). Therefore, the dynamic detection of sex hormone levels during radiotherapy will help to further explain this issue.

In the recent development of RT technology, the use of intensity-modulated radiation therapy and other innovations leads to a more accurate RT dose in the target area, with a smaller dose and less damage to the surrounding tissues. Thus, patients who were recently diagnosed with rectal cancer and received RT have a lower risk of second primary cancer than previous patients. With the use of more precise radiotherapy equipment, the changes in the type of second primary tumors may also be worth our forward-looking observation.

In our study, it can be seen that if divided into a young group and an elderly group of people who are 60 years of age or older, the proportion of young patients in the surgery combined with radiotherapy group is higher. However, after conducting a subgroup analysis, both the young and elderly groups have similar conclusions. Therefore, age may not be an important factor.

Our study has several limitations. Most importantly, with insufficient patient information, such as a family history of smoking and smoking history, selection bias is possibly observed. Moreover, some of the rectal cancer patients may have Lynch syndrome. Hence, they have an increased risk of developing secondary colorectal cancer and gastric cancer. The SEER database only provided limited treatment information on radiation and chemotherapy. Thus, we were unable to perform more analyses to investigate the association between second malignancy and the radiation regimen or the chemotherapeutic regimen. The information about tumor recurrence or treatment failure is not recorded or publicly available in the SEER database, so we are currently unable to conduct a statistical analysis on the tumor recurrence information.

## 5. Conclusions

In conclusion, this study demonstrates that patients with primary rectal cancer have a changed risk for the development of SPMs after RT. It would be beneficial to establish a risk profile for the development of SPMs. Potential risk factors can be identified by studying patients with SPMs.

## Figures and Tables

**Figure 1 medicina-59-01463-f001:**
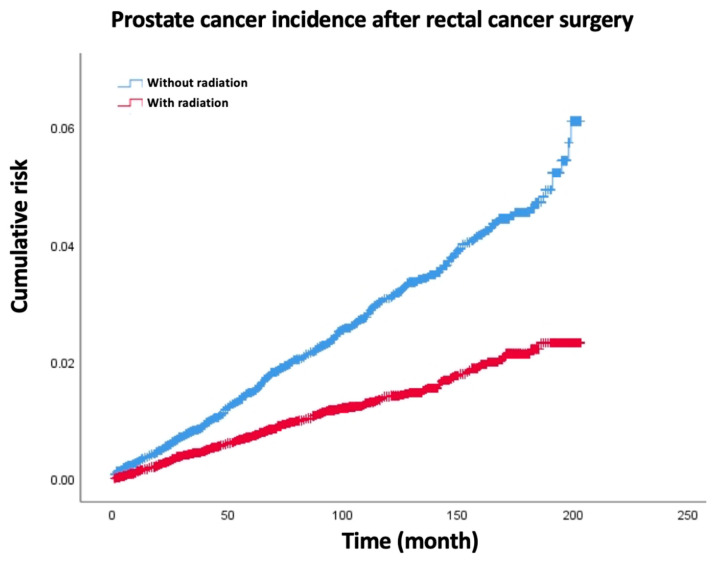
The secondary prostate cancer rate decreased significantly in radiation group compared with surgery-only group of rectal cancer patients.

**Figure 2 medicina-59-01463-f002:**
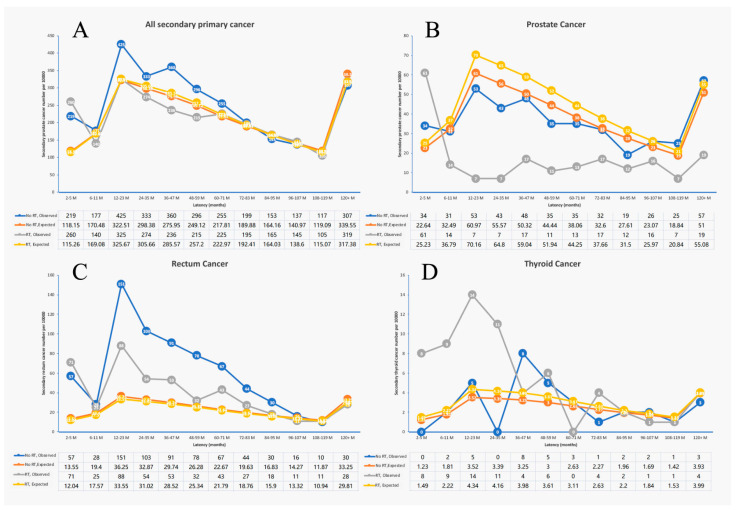
The second primary cancer patient number of different latency in surgery-only group and radiation group. (**A**) All second primary cancer patient number; (**B**) second primary prostate cancer patient number; (**C**) second primary rectum cancer patient number; (**D**) second primary thyroid cancer patient number.

**Table 1 medicina-59-01463-t001:** Descriptive characteristics of rectal cancer patients in 2000–2012.

	No RT, Surgery Only		RT + Surgery		Total	*p*-Value
Total	23,192	46%	26,769	54%	49,961	
Age						
<60	8017	38%	13,299	62%	21,316	<0.001
>60	15,175	53%	13,470	47%	28,645	
Gender						
Male	12,809	44%	16,615	56%	29,424	<0.001
Female	10,383	51%	10,154	49%	20,537	
Race						
White	19,380	47%	22,065	53%	41,445	<0.001
Black	1640	43%	2148	57%	3788	
Other	2172	46%	2556	54%	4728	
Diagnosis year of the primary cancer						
2000–2004	10,120	51%	9734	49%	19,854	<0.001
2005–2008	6813	45%	8496	55%	15,309	
2009–2012	6259	42%	8539	58%	14,798	
Tumor grade					0	
1	2702	61%	1719	39%	4421	<0.001
2	14,984	45%	18,614	55%	33,598	
3	2188	37%	3796	63%	5984	
4	155	39%	239	61%	394	
Unknown	3163	57%	2401	43%	5564	
Stage						
Localized	17,696	65%	9409	35%	27,105	<0.001
Regional	5472	24%	17,358	76%	22,830	
Unknown	24	92%	2	8%	26	
Chemo						
Yes	2189	47%	2481	53%	4670	<0.001
No/unknown	21,003	92%	1908	8%	22,911	

**Table 2 medicina-59-01463-t002:** The second malignancy pattern between radiation group and surgery-only group.

No RT, Surgery Only						RT + Surgery				
	Observed	Expected	O/E	Lower CI	Upper CI	Observed	Expected	O/E	Lower CI	Upper CI
All Sites	2978	2606.05	1.14 #	1.1	1.18	2604	2608.88	1	0.96	1.04
All Solid Tumors	2713	2292.03	1.18 #	1.14	1.23	2334	2311.65	1.01	0.97	1.05
Oral Cavity and Pharynx	54	61.43	0.88	0.66	1.15	64	67.66	0.95	0.73	1.21
Esophagus	33	30.43	1.08	0.75	1.52	39	31.85	1.22	0.87	1.67
Stomach	58	48.02	1.21	0.92	1.56	37	45.95	0.81	0.57	1.11
Colon, Rectum, and Anus	705	276.6	2.55 #	2.36	2.74	461	258.56	1.78 #	1.62	1.95
Liver	51	42.33	1.2	0.9	1.58	37	47.67	0.78	0.55	1.07
Pancreas	75	83.02	0.9	0.71	1.13	52	77.29	0.67 #	0.5	0.88
Lung and Bronchus	428	399.05	1.07	0.97	1.18	468	385.94	1.21 #	1.11	1.33
Soft Tissue including Heart	19	14.89	1.28	0.77	1.99	25	14.92	1.68 #	1.08	2.47
Melanoma of the Skin	90	118.99	0.76 #	0.61	0.93	99	123.93	0.80 #	0.65	0.97
Breast	243	254.93	0.95	0.84	1.08	193	232.25	0.83 #	0.72	0.96
Female Genital System	90	97.39	0.92	0.74	1.14	124	88.57	1.40 #	1.16	1.67
Cervix Uteri	8	7.94	1.01	0.43	1.98	6	8.08	0.74	0.27	1.62
Ovary	21	26.63	0.79	0.49	1.21	10	22.63	0.44 #	0.21	0.81
Male Genital System	443	463.19	0.96	0.87	1.05	213	530.14	0.40 #	0.35	0.46
Prostate	438	457.62	0.96	0.87	1.05	201	523.25	0.38 #	0.33	0.44
Urinary Bladder	155	162.82	0.95	0.81	1.11	177	156.09	1.13	0.97	1.31
Kidney	96	78.05	1.23	1	1.5	116	85	1.36 #	1.13	1.64
Endocrine System	35	32.53	1.08	0.75	1.5	73	37.75	1.93 #	1.52	2.43
Thyroid	32	30.1	1.06	0.73	1.5	64	35.1	1.82 #	1.4	2.33
All Lymphatic and Hematopoietic Diseases	200	238.45	0.84 #	0.73	0.96	217	231.75	0.94	0.82	1.07
Mesothelioma	9	8.11	1.11	0.51	2.11	6	7.64	0.79	0.29	1.71
Kaposi Sarcoma	4	1.54	2.59	0.71	6.63	0	1.64	0	0	2.25
Miscellaneous	52	62.03	0.84	0.63	1.1	40	53.24	0.75	0.54	1.02

# means *p* < 0.05.

**Table 3 medicina-59-01463-t003:** The second malignancy pattern between radiation group and surgery-only group in different latencies.

		No RT, Surgery Only					RT + Surgery				
Site	Latency	Observed	Expected	O/E	Lower CI	Upper CI	Observed	Expected	O/E	Lower CI	Upper CI
All Sites	2–5 m	219	118.15	1.85 #	1.62	2.12	260	115.26	2.26 #	1.99	2.55
All Sites	6–11 m	177	170.48	1.04	0.89	1.2	140	169.08	0.83 #	0.7	0.98
All Sites	12–23 m	425	322.51	1.32 #	1.2	1.45	325	325.67	1	0.89	1.11
All Sites	24–35 m	333	298.38	1.12	1	1.24	274	305.66	0.9	0.79	1.01
All Sites	36–47 m	360	275.95	1.30 #	1.17	1.45	236	285.57	0.83 #	0.72	0.94
All Sites	48–59 m	296	249.12	1.19 #	1.06	1.33	215	257.2	0.84 #	0.73	0.96
All Sites	60–71 m	255	217.81	1.17 #	1.03	1.32	225	222.97	1.01	0.88	1.15
All Sites	72–83 m	199	189.88	1.05	0.91	1.2	195	192.41	1.01	0.88	1.17
All Sites	84–95 m	153	164.16	0.93	0.79	1.09	165	164.03	1.01	0.86	1.17
All Sites	96–107 m	137	140.97	0.97	0.82	1.15	145	138.6	1.05	0.88	1.23
All Sites	108–119 m	117	119.09	0.98	0.81	1.18	105	115.07	0.91	0.75	1.1
All Sites	120+ m	307	339.55	0.9	0.81	1.01	319	317.38	1.01	0.9	1.12
All Sites	Total	2978	2606.05	1.14 #	1.1	1.18	2604	2608.88	1	0.96	1.04
Colon, Rectum, and Anus	2–5 m	57	13.55	4.21 #	3.19	5.45	71	12.04	5.90 #	4.61	7.44
Colon, Rectum, and Anus	6–11 m	28	19.4	1.44	0.96	2.09	25	17.57	1.42	0.92	2.1
Colon, Rectum, and Anus	12–23 m	151	36.25	4.17 #	3.53	4.89	88	33.55	2.62 #	2.1	3.23
Colon, Rectum, and Anus	24–35 m	103	32.87	3.13 #	2.56	3.8	54	31.02	1.74 #	1.31	2.27
Colon, Rectum, and Anus	36–47 m	91	29.74	3.06 #	2.46	3.76	53	28.52	1.86 #	1.39	2.43
Colon, Rectum, and Anus	48–59 m	78	26.28	2.97 #	2.35	3.7	32	25.34	1.26	0.86	1.78
Colon, Rectum, and Anus	60–71 m	67	22.67	2.96 #	2.29	3.75	43	21.79	1.97 #	1.43	2.66
Colon, Rectum, and Anus	72–83 m	44	19.63	2.24 #	1.63	3.01	27	18.76	1.44	0.95	2.09
Colon, Rectum, and Anus	84–95 m	30	16.83	1.78 #	1.2	2.55	18	15.9	1.13	0.67	1.79
Colon, Rectum, and Anus	96–107 m	16	14.27	1.12	0.64	1.82	11	13.32	0.83	0.41	1.48
Colon, Rectum, and Anus	108–119 m	10	11.87	0.84	0.4	1.55	11	10.94	1.01	0.5	1.8
Colon, Rectum, and Anus	120+ m	30	33.25	0.9	0.61	1.29	28	29.81	0.94	0.62	1.36
Colon, Rectum, and Anus	Total	705	276.6	2.55 #	2.36	2.74	461	258.56	1.78 #	1.62	1.95
Prostate	2–5 m	34	22.64	1.50 #	1.04	2.1	61	25.23	2.42 #	1.85	3.11
Prostate	6–11 m	31	32.49	0.95	0.65	1.35	14	36.79	0.38 #	0.21	0.64
Prostate	12–23 m	53	60.97	0.87	0.65	1.14	7	70.16	0.10 #	0.04	0.21
Prostate	24–35 m	43	55.57	0.77	0.56	1.04	7	64.8	0.11 #	0.04	0.22
Prostate	36–47 m	48	50.32	0.95	0.7	1.26	17	59.04	0.29 #	0.17	0.46
Prostate	48–59 m	35	44.44	0.79	0.55	1.1	11	51.94	0.21 #	0.11	0.38
Prostate	60–71 m	35	38.06	0.92	0.64	1.28	13	44.25	0.29 #	0.16	0.5
Prostate	72–83 m	32	32.6	0.98	0.67	1.39	17	37.66	0.45 #	0.26	0.72
Prostate	84–95 m	19	27.61	0.69	0.41	1.07	12	31.5	0.38 #	0.2	0.67
Prostate	96–107 m	26	23.07	1.13	0.74	1.65	16	25.97	0.62	0.35	1
Prostate	108–119 m	25	18.84	1.33	0.86	1.96	7	20.84	0.34 #	0.14	0.69
Prostate	120+ m	57	51	1.12	0.85	1.45	19	55.08	0.34 #	0.21	0.54
Prostate	Total	438	457.62	0.96	0.87	1.05	201	523.25	0.38 #	0.33	0.44
Thyroid	2–5 m	0	1.23	0	0	3	8	1.49	5.35 #	2.31	10.55
Thyroid	6–11 m	2	1.81	1.1	0.13	3.99	9	2.22	4.06 #	1.86	7.7
Thyroid	12–23 m	5	3.52	1.42	0.46	3.31	14	4.34	3.23 #	1.76	5.42
Thyroid	24–35 m	0	3.39	0	0	1.09	11	4.16	2.64 #	1.32	4.73
Thyroid	36–47 m	8	3.25	2.46 #	1.06	4.85	4	3.98	1.01	0.27	2.57
Thyroid	48–59 m	5	3	1.67	0.54	3.89	6	3.61	1.66	0.61	3.62
Thyroid	60–71 m	3	2.63	1.14	0.24	3.34	0	3.11	0	0	1.19
Thyroid	72–83 m	1	2.27	0.44	0.01	2.45	4	2.63	1.52	0.42	3.9
Thyroid	84–95 m	2	1.96	1.02	0.12	3.69	2	2.2	0.91	0.11	3.28
Thyroid	96–107 m	2	1.69	1.19	0.14	4.28	1	1.84	0.54	0.01	3.02
Thyroid	108–119 m	1	1.42	0.7	0.02	3.91	1	1.53	0.65	0.02	3.65
Thyroid	120+ m	3	3.93	0.76	0.16	2.23	4	3.99	1	0.27	2.57
Thyroid	Total	32	30.1	1.06	0.73	1.5	64	35.1	1.82 #	1.4	2.33

# means *p* < 0.05.

**Table 4 medicina-59-01463-t004:** Descriptive characteristics of prostate cancer after rectal cancer.

	No RT, Surgery Only		RT + Surgery		Total	*p*-Value
Total	521		275		796	
Age						0.675
<60	61	11.70%	35	12.70%	96	
>60	460	88.30%	240	87.30%	700	
Race						0.904
White	427	82.00%	223	81.10%	650	
Black	64	12.30%	34	12.40%	98	
Other	30	5.80%	18	6.50%	48	
Tumor grade						0.331
1	31	6.00%	13	4.70%	44	
2	221	42.40%	121	44.00%	342	
3	221	42.40%	104	37.80%	325	
4	1	0.20%	1	0.40%	2	
Unknown	47	9.00%	36	13.10%	83	
Stage						0.044
Localized	418	80.20%	200	72.70%	618	
Regional	38	7.30%	31	11.30%	69	
Distant/unknown	65	12.50%	44	16.00%	109	
Chemo						0.001
Yes	3	0.60%	10	3.60%	13	
No/unknown	518	99.40%	265	964%	783	
Median latency months	51 (21.75, 95)		47 (4.5, 86)			

## Data Availability

Limited Use Agreement for Surveillance, Epidemiology, and End Results (SEER) Program (https://seer.cancer.gov, accessed on 1 May 2021) SEER*Stat Database: released in April 2019, based on the November 2018 submission. The data can be used publicly.

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
