# Peer review of "Radiation Therapy Changed the Second Malignancy Pattern in Rectal Cancer Survivors"

_medicina, 2023, doi:10.3390/medicina59081463_

Round 1
Reviewer 1 Report
In the article "Radiation therapy changed the secondary malignancy pattern in rectal cancer survivors" Ye et al. proposes an analysis of the risk of secondary malignancies in rectal cancer survival on a very large number of cases. In fact, the article identifies a different pattern for cases that have benefited from radiotherapy in the past. If a possible explanation is offered for the reduction of the risk of localized prostate cancer, in the case of thyroid cancer (more frequently in the group treated with radiotherapy) it is not discussed (is it related to secondary neutrons? is it possible to spread the lower doses in large volumes through IMRT and VMAT techniques?) Also an expansion and a higher number of references for discussions is necessary.
Author Response
Reply: Thank you very much for your suggestions, as the information of radiation techniques were not recorded in SEER database, we can not do further analysis to know whether IMRT and VMAT technique maybe more helpful. And we have added some discussion according to your suggestions.
Reviewer 2 Report
The manuscript uses SEER data to investigate whether secondary cancers have a different pattern in rectal cancer patients treated with radiotherapy. Overall, this is a clear rationale and worthwhile to understand. I have some comments to improve the manuscript:
- the introduction is very short, and needs expanding. The authors should provide full context for their study - if the hypothesis is that ionizing radiation leads to secondary cancer, then be clear and provide background evidence for this. What mechanistic understanding of this is already known? Provide details on radiotherapy response in rectal cancer - its important to understand why it is used if this study is demonstrating associated risk. Expand on what conclusions are different between different previous studies, provide a rationale for why this study is necessary otherwise the originality/novelty is not really clear.
- I am interested in the age different between RT/non-RT group. Could the pattern differ based on the choice to use RT in a younger, less vulnerable population? Given that age is a primary risk factor for many of the types of secondary cancer shown, it is surprising that little consideration is given to age as a confounding variable. At least needs discussing.
- Can the authors clarify for a non-clinician, secondary cancer of the colon/rectum/anus - would this be classed as recurrence, and represent failure of treatment vs increased risk of inducing secondary tumour? This is an important mechanistic distinction, and again relates to the fact that the context/rationale isn't clear enough at the start.
English generally fine, some minor errors.
Author Response
Reply: 1. Thank you very much for your suggestions, we have expanded the introduction part according to your suggestions on Page 2 lines 46-52.
2. Indeed, in our study, it can be seen that if we take the age of 60 as the boundary, the proportion of young patients in the surgery combined with radiotherapy group is indeed higher. However, the results of this article are relatively consistent, and the situation is similar for both young and elderly patients. We have also supplemented the discussion section of the article based on your suggestion on Page 9 lines 204-207.
3. The Information on whether there is recurrence or treatment failure is not recorded or publicly available in the SEER database and cannot be obtained. We can only obtain the second primary cancer, but not the recurrence situation. Thus currently this information cannot be supplemented and analyzed. Based on your suggestion in the discussion section, we have also discussed the shortcomings in this area on Page 9 lines 215-217.
Reviewer 3 Report
This present article by Ye et al, reported Radiation therapy changed the secondary malignancy pattern in rectal cancer. They reported radiotherapy might change the secondary malignancy pattern in rectal cancer survivors, the risk of prostate cancer decreased, and the risk of thyroid cancer increased most significantly. It is important to note that, patients who received RT had a lower risk of developing ovarian, prostate, and breast cancer and a higher risk of developing thyroid, lung, and bronchial cancer than patients who did not receive RT. Authors also agree with their limitations in the present study by not considering other factors such as family history of smoking and smoking history, selection bias is possibly observed. Moreover, some of the rectal cancer patients may have Lynch syndrome. Overall this article may be helpful in establishing a risk profile for the development of SPMs. Therefore, I do principally support this work for publication.
Minor
The graph presented in figure 2, the axis label should be legible.
Minor language corrections required. Overall the article is well written.
Author Response
Reply: Thank you very much for your suggestions, we have changed the axis labels of figure 2 according to your suggestions in the revision.
Round 2
Reviewer 1 Report
Even if the results are very interesting, assuming that I am not a radiation oncologist and I know very little data about the risk of thyroid cancer associated with radiation, I would like to see some hypotheses and correlations from the literature. Is it possible that some cases were treated with energies that favor the release of fast neutrons? A subgroup analysis to identify factors associated with thyroid cancer after pelvic irradiation should be specified as a future project after this study. Also, the authors could issue/justify a hypothesis about the increased incidence of regional and metastatic prostate cancers, being a change in the pattern, not just a reduction of the incidence through local sterilization of tumor cells. I appreciate stating the limitations of the study but the topic can be developed. For a non-expert in the field, it would be interesting to find out how the radiotherapy technique or chemotherapy regime could influence the risk of prostate cancer. A simple phrase like "Cigarette smoking may increase the risk of prostate cancer by affecting circulating hormone levels or through exposure to carcinogens" (Plaskon et al., 2003) could provide an explanation for the implications of the limitations of the study, prostate cancer not being a notorious cancer associated with smoking.
Author Response
Thanks for your suggestion! We have added our hypothesis for the effects of radiotherapy on the risk of prostate cancer in the discussion on Page 8 lines 198-204.